# Synthesis and Applications of Nitrogen-Containing Heterocycles as Antiviral Agents

**DOI:** 10.3390/molecules27092700

**Published:** 2022-04-22

**Authors:** Tuyen N. Tran, Maged Henary

**Affiliations:** 1Department of Chemistry, Georgia State University, 100 Piedmont Avenue SE, Atlanta, GA 30303, USA; ttran148@student.gsu.edu; 2Center of Diagnostics and Therapeutics, Department of Chemistry, Georgia State University, 100 Piedmont Avenue SE, Atlanta, GA 30303, USA

**Keywords:** nitrogen-containing heterocycles, synthesis, antiviral agents, viruses, COVID-19, inhibition

## Abstract

Viruses have been a long-term source of infectious diseases that can lead to large-scale infections and massive deaths. Especially with the recent highly contagious coronavirus (COVID-19), antiviral drugs were developed nonstop to deal with the emergence of new viruses and subject to drug resistance. Nitrogen-containing heterocycles have compatible structures and properties with exceptional biological activity for the drug design of antiviral agents. They provided a broad spectrum of interference against viral infection at various stages, from blocking early viral entry to disrupting the viral genome replication process by targeting different enzymes and proteins of viruses. This review focused on the synthesis and application of antiviral agents derived from various nitrogen-containing heterocycles, such as indole, pyrrole, pyrimidine, pyrazole, and quinoline, within the last ten years. The synthesized scaffolds target HIV, HCV/HBV, VZV/HSV, SARS-CoV, COVID-19, and influenza viruses.

## 1. Introduction

In recent years, outbreaks of infectious viral diseases have been increasing unexpectedly and costing millions of human lives. Despite many developed vaccines and therapeutics for prevention and treatment, viruses have continuously evolved and re-emerged to threaten public health, social relations, and economic stability.

In the early 1980s, the human immunodeficiency virus/acquired immunodeficiency syndrome (HIV/AIDS) infected 79 million people, with 39 million deaths over three decades [1]. The influenza virus series also have a long-term effect on public health. The seasonal influenza viruses infect 2–5 million people and kill 250,000–500,000 people worldwide per year [2]. Other influenza viruses also have caused occasional pandemics throughout history. The H1N1 influenza pandemic of 1918 cost approximately 40 million lives. H2N2 caused another epidemic in 1957, while H3N2 struck in 1968, and H1N1 again in 2009 [2]. During the early 21st century, the severe acute respiratory syndrome coronavirus (SARS-CoV) from a zoonotic source started in Guangdong Province, China. It spread through the global community, causing about 8000 infections and 800 deaths worldwide [3]. Middle East respiratory syndrome coronavirus (MERS-CoV) emerged in 2012 and caused 2029 infections and 704 deaths in 27 countries [4]. The novel coronavirus (SARS-CoV-2) has caused a worldwide pandemic with more than 99 million infections and 2 million deaths within the first twelve months, starting in December 2019 in Wuhan, China. Although some antiviral drugs and vaccines are available for certain viruses, it is necessary to continuously develop new drugs and methods for drug-resistant viruses and viruses newly evolved from mutations.

The most promising antiviral drugs are small organic molecules that can target specific parts of the viruses and interfere with different stages of the viral life cycle [4]. Various strategies have been applied to target different viruses effectively. For example, for SARS-CoV, small molecules were used to target protease activity and inhibit viral replication. For anti-HIV/AIDS agents, viral glycoproteins were targeted to hinder their interaction with the receptors on cell surfaces, which then activate the virus’s endocytosis into the cell [4]. Among many chemical scaffolds, nitrogen-containing heterocyclic small compounds have been exploited extensively due to their broad range of applications in biological and pharmacological activities. The nitrogen-containing heterocyclic bases have high versatility in synthesis with different moieties. Those heterocyclic backbones have rigid aromatic structures that can be incorporated into the binding pockets and provide various molecular interactions, such as ionic bonding, hydrogen bonding, hydrophobic interaction, non-covalent bonding, etc., for ligand binding with receptor proteins [5,6]. The interaction can mimic specific properties that can effectively inhibit the activities of various biological enzymes and components that play vital roles in the development of viral infections [7,8]. Previous reviews focused on single types of nitrogen-containing heterocycles such as indole and/or imidazothiazole derivatives in designing antiviral agents [7,9]. Other reviews covered broader details on various biological applications such as antimicrobial, anti-inflammatory, anti-tubercular, anti-depressant, and anti-cancer activities [10,11,12]. Our current review focuses specifically on the common nitrogen-containing heterocycles including indoles, pyrroles, pyrimidines, pyrazoles, and quinolines that have been applied in drug design for antiviral purposes during the past ten years. These heterocycles are significant backbones of pharmaceutical products with exceptional biological activity to interfere with various viral infections [8]. While the indole cores are known for their wide existence in natural products with biological activity [7], the properties of pyrrole can be expanded for chemical design and are suitable for biological systems [13]. Pyrimidine derivatives have widespread therapeutic applications, as they are essential building blocks of nucleic acids in DNA and RNA [14]. Pyrazoles can be fused with other heterocycles to extend their active biological potential [15]. Quinoline derivatives have versatile chemical properties for synthesis and biological activities [16]. Our review discusses the synthesis and biological characteristics of nitrogen-containing heterocycle derivatives in detail. Further modifications and functions of nitrogen heterocycles are introduced along with various antiviral purposes.

## 2. Indole

Indole derivatives are one of the well-known scaffolds in drug discovery that can inhibit a wide diversity of enzymes by binding with potential ligands [7]. The first indole was prepared using Fischer indole synthesis, which was reported in 1866. Nowadays, most indole synthesis pathways start with incorporating a benzene ring with additional factors that stabilize the formation with the fused pyrrole ring. Indole is an aromatic ring with ten π electrons and a lone pair of nitrogens. The lone pair is not available for protonation but is involved in the delocalization of the indole conjugated system. The protonation and other electrophilic substitutions mostly occur at the C-3 position, which has the highest electron density and is most thermodynamically stable for modifying antiviral agents [17]. 

Hassam et al. designed new scaffolds of HIV non-nucleoside reverse transcriptase (RT) inhibitor with the addition of a cyclopropyl group at the C-3 of the indole base [18]. The RT enzyme plays a crucial role in the reverse transcription of viral RNA to single-stranded DNA in host cells. The binding of specific cyclopropyl chemical moiety to the RT pocket can significantly inhibit the enzyme’s function by improving electrostatic interaction with the hydrophobic pocket [18]. As shown in Figure 1, the ester-substituted indole **1** reacted with benzoyl chloride for the electrophilic substitution at the C-3 position. Tert-butyloxycarbonyl (Boc) was added to intermediate **2** to prevent reactions at the amine before further modifying cyclopropyl moiety. Intermediate **3** underwent a Wittig reaction with methyl triphenylphosphonium ylide reagent to convert ketone to alkene **4**, which then reacted with diiodomethane and zinc alloy for cyclopropyl addition to furnish product **5** [18]. 

As shown in Figure 1, in the in vitro phenotypic assay, compounds **5a–c** showed high inhibitory activity (IC_50_), with low cytotoxicity (CC_50_) in **5a** and **5c**. In the modeling study, **5a** was better at accommodating the binding site. The indole NH and ester moiety was incorporated for hydrogen bonding in the reverse transcriptase’s allosteric site at Lys 101. The ester group associates well in the binding site with the ethyl directed out of the site. Modification of the ethyl group decreased the potency of the compound. The cyclopropyl moiety was implemented to bind with Val 179 binding pocket, a small, hydrophobic cleft located near the catalytic site. The aromatic ring of R_2_ was favorable for the interaction with Tyr 188 and Trp 299. The modification at R_2_ from phenyl to 2-thiophenyl resulted in a slightly better inhibitory value but increased toxicity in cells. Increasing the size of the halogen at R_3_ also enhanced IC_50_ and CC_50_ values [18].

Another study of antiviral agents with substitution at C-3 of indole was utilized to target the hepatitis C virus (HCV). HCV infection can lead to acute or chronic liver diseases, such as hepatocellular carcinoma and liver cirrhosis. Specific treatments have been approved for clinical applications. However, those treatments possessed low effectiveness in the patient population with additional side effects [19]. Han et al. [20] introduced the small molecules of *N*-protected indole scaffolds (NINS) to inhibit HCV. As shown in Figure 2, scaffold **6** reacted with the racemic epibromohydrin to form *N*-protection derivative **7** with two enantiomers. The substitution of NH was further extended by ring-opening of epoxide with the nucleophilic amine chain to afford **8**. 

In the structure–activity relationship (SAR) study, multiple modifications of the phenyl ring at R_1_ were analyzed. As shown in Table 1, *p*-F-Ph and *m*-F-Ph improved inhibitory potency, but *o*-F-Ph reduced anti-HCV activity. The *m*-F-Ph exhibited lower cytotoxicity compared to *p*-F-Ph. The (R)-enantiomer can possess better anti-HCV potency and less cytotoxicity than its corresponding (S)-enantiomer of both *m*-F-Ph and *p*-F-Ph. The results indicated that the position of substitution and chirality has a high impact on their inhibitory effect. The scaffold was also proved to inhibit viral entry into the viral cycle rather than interfere in any viral RNA replication in host cells with a mechanism of action (MoA) study.

The varicella-zoster virus (VZV) infection can cause acute varicella and herpes zoster, leading to various disease complications in the central nervous system from latent viruses [21]. Most approved drugs for VZV-associated treatments were nucleosides that competitively inhibit the viral DNA polymerase and interfere with DNA replication. However, those nucleosides implied multiple drug resistance and produced low efficacy in the anti-VZV virus. Mussela et al. [22] synthesized a family of indole derivatives as non-nucleoside antivirals to mimic deoxythymidine. The compounds can inhibit thymidine kinase (TK) by interfering with the series phosphorylation of thymidine triphosphates, which can be used as a complementary base in the replication process [23]. As shown in Figure 3, the 3-(2-bromoethyl)-*1H*-indole **9** was reacted with methyl iodide for *N*-alkylation. The nucleophilic displacement of bromine in **10** was conducted with chain-linked amine and used palladium as a catalyst under microwave conditions to obtain **11**, which was then treated with acetyl chloride under the basic condition to yield the final compound **12**. 

Compound **12** has low cytotoxicity in cells and good inhibitory activity at low EC_50_ due to the alkylation of NH and acylation of the amine group in C-3 substitution [22]. This tryptamine derivative has lower potency than reference drugs Acyclovir and Brivudine but displayed similar inhibiting activity against TK in anti-VZV mechanisms. Additionally, the derivative **12** selectively targeted VZV strains (OKA, 07-1, and YS-R) only when tested with other members in the same family of Herpesviridae (HMCV, HSV-1, and HSV-2) and various RNA virus strains such as HIV and influenza, as shown in Table 2. Compound **12** can be a leading compound for further investigation in inhibiting VZV specifically. 

Other synthesized potential antiviral agents have substitution at C-2 of indole, the second most reactive site for electrophiles. For targeting SARS-CoV viruses, Thanigaimalai et al. [24] discovered compounds to target the chymotrypsin-like protease (CL^pro^), which plays a vital role in cleaving polyproteins to produce functional proteins directly involved in viral replication and transcription [25]. The analogs were peptidomimetic covalent inhibitors that can mimic the substrate of CoV 3CL^pro^. As shown in Figure 4, to synthesize compound **17**, the peptidomimetic chain was added to the commercially available indole derivative **13**, through peptide coupling with the carboxyl group at C2 to form the peptide **14**. For the other intermediate, γ-lactam acid **15** coupled with *N, O*-dimethylhydroxylamine to form substituted amide from the carboxylic end using Weinreb–Nahm synthesis. The amide continuously reacted with benzothiazole to create **16**, which then deprotected and coupled with the peptide **14** in the presence of peptide coupling reagent to furnish compound **17** [24].

The synthetic inhibitor **17** has four main features that are suitable for different pockets in the active site of CoV CL^pro^. The compounds included ketone to target cysteine residue’s thiol (Cys 145) in the S1″ pocket and (S)-γ-lactam ring for the S1 pocket. The hydrophobic leucine was used for the S2 position [24,26]. In another study, the leucine was replaced by the π conjugated system and hydrophobic interaction of the aryl or cyclohexyl group to enhance access of the S2 pocket to target SARS-CoV-2 [27]. Based on its SAR study, compound **17** exhibited great inhibitory (Ki = 0.006 μM or IC_50_ = 0.74 μM) activity with the methoxy at C-4 of the indole. Indole possessed the best inhibitory effect among other heterocycles due to its NH, which can provide hydrogen bonding interaction with Gln 166, as shown in Figure 2 from the molecular docking study. 

## 3. Pyrrole

Pyrrole is a heterocyclic aromatic five-membered ring that was first observed in coal tar and bone oil in 1834. Many investigations reported pyrrole as an integral part of different natural compounds. The delocalization of the lone pair of electrons from the nitrogen atom provided additional stabilization of the ring. In total, six π electrons delocalized over the five-membered ring formed the isoelectronic system and allowed electrophilic attack in different reactions [28]. 

Curreli et al. [29] have developed drugs to block HIV-1 envelope glycoprotein (gp120) from binding the receptor CD4 of the host cells and prevent the entry of viral RNA [30]. The scaffolds were designed to mimic receptor CD4 and act as HIV-1 entry antagonists. Intermediates **18** and **19** were synthesized and coupled to yield compound **20**, as shown in Figure 5 [28,31]. The deprotonation of amine followed to afford product **21**. Intermediate **19** can be synthesized from R- and S-isomer imines that derived isomers in compounds **20** and **21** [29]. 

Based on X-ray crystal structure analysis, pyrrole allowed a potential hydrogen bond of NH with the residue Asn425 of gp120. Methylation of NH lost an H-bond donor atom and increased steric hindrance, which interferes with binding capability. The scaffold loses its antiviral activity when replacing the pyrrole ring with imidazole. According to Table 3, the compounds exhibited high antiviral potency with low cytotoxicity and good selectivity for HIV-1 gp120. The methyl substitution at R_2_ of **21a** enhanced antiviral potency, while the addition of fluoro at R_3_ of **21b** showed similar activity but improved metabolic stability. Because **(R) 21a** (NBD-14088) and **(S) 21b** (NBD-14107) have better selectivity, they were used to measure HIV-1 entry antagonist properties with cell-to-cell fusion inhibition assay and infectivity in cells, as shown in Figure 3. Even though **(R) 21a** and **(S) 21b** required higher IC_50_ compared to NBD-556 (HIV-1 entry agonist) and NBD-11021 (HIV-1 entry antagonist), negative and positive controls, respectively, two synthesized compounds still exhibited antagonist properties to reduce infections. The compounds also can inhibit HIV-1 reverse transcriptase from converting viral RNA into complementary DNA in hosts [29]. 

Herpes simplex virus (HSV) is another virus in the family of α-herpesviruses with VZV that can cause common, self-resolving diseases of skin or mucosa, such as herpes labialis and other infectious diseases. However, HSV and VZV possess differences in the route of infection, spread pathway, and range of hosts [32]. Due to the viral strains highly resistant to the nucleoside treatments, non-nucleoside compounds were synthesized to inhibit thymidine kinase (TK) from early DNA replication. According to Hilmy et al. [33], the pyrrole analogs **23a**–**d** were synthesized using different substituted 2-amino-3-cyano-1,5-diarylpyrroles to react with various aryl aldehydes under the basic condition with phosphorous pentoxide, as shown in Figure 6.

The analogs were compared with Acyclovir (ACV), a standard drug used to treat HSV, for anti-HSV activity and cytotoxicity. As shown in Table 4, all new compounds exhibited a high percentage of reduction (94–99%) in the number of virus plaques. Compounds **23a** (99%) and **23d** (97%) even had better results than ACV due to their similarity at substituted *N* of pyrrole with 4-methoxyphenyl. Compound **23a** had the highest activity with the 4-methoxybenzylideneamino at position C-2 of pyrrole. The synthesized compounds had better docking scores than ACV, indicating that they had better ligand–receptor interaction in the TK active site. Comparing compounds **23a** and **23d** in Figure 4, they had different interactions in the binding pocket. The two oxygens of the methoxy group in **23a** had hydrogen bonding with Lys62 and Tyr132. Another hydrogen bond was formed between the cyano nitrogen and Arg222. However, for **23d**, the methoxy group formed two hydrogen bonds with Arg176 and Tyr101. Other hydrogen bonds were also established from the nitrogens of the compounds. Both also formed hydrophobic and van der Waals interactions with other amino acids. However, **23d** interacted with crucial amino acids that actively contribute to the function of TK; hence, **23d** showed better inhibitory effect compared to **23a**. Additionally, compounds **27b** and **27c** might have different anti-HSV activity mechanisms rather than targeting TK [33].

Lin et al. [34] introduced a class of anti-influenza agents targeting the viral nucleoprotein (NP), a binding protein that contributes to the transcription and packaging processes. The synthesized pyrimido-pyrrolo-quinoxalinedione analogs were aimed to inhibit the synthesis of NP and interrupt viral replication [35]. As shown in Figure 7, substituted pyrimidinedione **24** was methylated at the two atoms of nitrogenusing dimethyl sulfate, followed by Friedel–Crafts acylation with benzyl chloride to furnish compound **26**. The synthesis was continued with the addition of bromine and the formation of fused pyrrole **28** from the reaction of **27** with 2-amino-2-methylpropan-1-ol. The annulation of intermediate **28** with *F*- substituted aryl aldehyde produced the final product, compound **29**.

The inhibiting effect of compound **29** (PPQ-581) was compared with that of the nucleozin 3061, a potent antagonist of NP. Both have a similar trend of effectively inhibiting nucleoprotein (NP) synthesis shortly after infection. Treatment with **29** and nucleozin from 3–8 h post-infection partially inhibits NP synthesis. The docking study of compound **29** in the influenza A nucleoprotein is shown in Figure 5. The oxygen of ketone formed hydrogen bonding with S377, which was a crucial binding area for compound **29.** The mutation of the S377 sidechain can result in the loss of the anti-influenza activity of compound **29.** Compound **29** also inhibited the influenza RNA-dependent RNA polymerase (RdRP) activity of nucleozin-resistant influenza strains.

## 4. Pyrimidine

Pyrimidine is a heterocycle composed of a six-membered ring with two nitrogens at positions 1 and 3. The first pyrimidine derivative was synthesized in 1818 by Gaspare Brugnatelli. Pyrimidines were continuously developed for applications in biological systems, as their nitrogen bases were associated with DNA and RNA. The two nitrogens in pyrimidines are electron withdrawers, leaving specific carbons with electron deficiency [36]. Pyrimidine has also been fused with other heterocycles to improve its applicable spectrum of biological activity [37].

Currently, HIV/AIDS is still causing a major global health issue. The treatments for HIV/AIDS require a long-term commitment to target different stages of viral DNA replication. Malancona et al. [38] introduced the 5,6-dihydroxypyrimidine scaffolds to target the HIV nucleocapsid (NC), which is a multifunctional protein that plays crucial roles in binding amino acids for their sequence-specific binding in reverse transcription. NC is also a nucleic acid chaperone that can manipulate nucleic acid structures for their thermodynamically stable conformations [39]. As shown in Figure 8, the central dihydroxypyrimidine cores were obtained by reacting substituted 2-pyridinecarbonitrile with hydroxylamine hydrochloride to form intermediate **31**. The synthesis was continued with the reaction of compound **31** with dimethyl acetylenedicarboxylate (DMAD) for thermal cyclization to produce **32**, which coupled with (R)-1-cyclohexylethanamine to furnish **33a**–**33d**.

The 2-(pyridin-2-yl) substituent at R1 was essential for the inhibitory activity of the analogs. The amide moiety (*N*-cyclohexylmethyl) targeted the hydrophobic binding site of the protein. As shown in Table 5, further modification with 5-methoxy, 5-chlorine at the pyridine group, and quinoline resulted in **33b**–**33d** with 40–80 folds of improved potency and higher SI values (25–75) than without any modification of 2-(pyridin-2-yl) substituent. As shown in Figure 6, the molecular docking of compound **33c** in the NC demonstrated that 2-(pyridin-2-yl) substituent has π-stacking interaction with Trp37, while the dihydroxypyrimidine formed H-bonds with Gly35, Met46, and Gln45. The amide moiety interacted with the hydrophobic pocket, including Phe16, Ala25, Trp37, and Met46. Other in vitro and in vivo tests against multiple drug-resistant HIV-1 strains were also analyzed in the study. The analogs were also active in different HIV-1 resistant strains, with high oral bioavailability and excellent in vitro metabolic stability in rat and human samples and half-life in the in vivo study [38].

Mohamed et al. [40] reported the synthesis of non-nucleoside antiviral agents for hepatitis C virus (HCV), which has the highest infection rate in Egypt and causes 350,000 deaths worldwide annually. The HCV NS5B polymerase functioned as RNA-dependent RNA polymerase (RdRp) and enabled the catalyzation of viral genome synthesis [41]. The synthesis of pyrrolopyrimidine derivative **39** as a non-nucleoside purine scaffold to inhibit HCC NS5B polymerase is summarized in Figure 9. The benzoin **34** was condensed with aryl amine to form intermediate **35**, which then reacted with malononitrile to produce pyrrole derivative **36** via Dakin West reaction. The 2-aminopyrrole-3-carbonitrile **36** coupled with formic acid to furnish associated pyrrolopyrimidine derivative **37**. The carbonyl was replaced by chlorine at C-4 pyrimidine, resulting in intermediate **38** for further modification at this position. Aryl amine was introduced to produce 4-aryl amino derivative **39**. 

Compound **39** exhibited the highest antiviral activity in the study. The arylamino group (‘‘) enhanced the toxicity of the scaffold in HCV genotype 4a cells. In the molecular docking, it was found that **39** bound strongly with Mg^2+^ in the docked site. Other measurements in the docking system proved that **39** was stable in the binding pocket and improved binding affinity. As shown in Figure 7, the carbonyl C=O of Gln446 in the HCV RdRp catalytic site can form a hydrogen bond with aniline NH. At the same time, the NH_2_ interacted with nitrogen in the pyrimidine ring to enhance the binding of compound **39** to the inhibition site.

Pyrimidine derivatives were also employed in the study of anti-HSV activity. Currently, Acylclovir (Figure 8) is one of the drugs used to treat the herpes simplex virus (HSV). However, in some cases, the viruses can resist the drug; hence, new agents are continuously developed and tested for antiviral activity. Mohamed et al. [42] reported the synthesis of pyrimidine derivatives as anti-HSV agents. As shown in Figure 10, acetophenone **40** was coupled with 3,4-dimethoxybenzaldehyde via aldol condensation to afford chalcone **41**, which was reacted with hydrogen peroxide to form epoxide **42**. The condensation of compound **42** with thiourea resulted in the cyclic formation of compound **43**, which could either react with 3-chloroacetylacetone to produce compound **44** or be condensed with 2-bromopropionic acid to form methylthiazolo compound **45**. 

For the antiviral activity of compounds **44** and **45**, a plaque reduction assay was conducted using concentrations of 2 and 5 μg/mL, and the viral count was recorded after adding the compounds. As shown in Table 6, compounds **44** and **45** showed a slightly higher percentage of inhibition with the 5 μg/mL concentration. Compound **44** was more effective than compound **45,** achieving 100% inhibition with 5 μg/mL in the test sample. As shown in Figure 8, both compounds showed a higher percentage of inhibition at much lower concentrations compared to Acyclovir^®^. For the antiviral mechanism, **44** and **45** showed viricidal activity that possibly altered viral epitopes to inhibit binding to cells. Besides viricidal activity, compound **44** also has the potential to interfere with the replication processes of HSV. The two compounds are highly promising as potential new anti-HSV agents. 

During the current COVID-19 pandemic, Remdesivir^®^ (Figure 9) was approved for use in the treatment of patients with confirmed SARS-CoV-2 infection. Remdesivir^®^ is an adenosine nucleoside prodrug that exhibits a broad antiviral spectrum. Due to the rapid emergence of coronavirus infections that threaten millions of people in the global community, a safe, approved drug like Remdesivir^®^ was temporarily employed for treatment. It showed in vitro antiviral activity with animal and human coronaviruses, including SARS-CoV-2. As shown in Figure 10, Remdesivir^®^ can effectively inhibit viral infection at a low concentration (EC_50_ = 0.77 μM) with a high SI value (129.87 μM). The antiviral activity of Remdesivir^®^ was confirmed with an immunofluorescence assay. The results indicated complete viral reduction, as the viral nucleoproteins could not be observed with a 3.70 μM concentration at 48 h post-infection [43].

Remdesivir^®^ was originally targeted to inhibit RNA-dependent RNA polymerase (RdRp), which has adenosine triphosphate (ATP) as the main substrate. Zhang et al. [44] reported the homology modeling of NSP12 (the RdRp complex with multiple nonstructural protein units) of SARS-CoV-2 RdRp (SARS-CoV-2 NSP12). When accumulated in cells, Remdesivir^®^ was hydrolyzed and coupled with triphosphate (RemTP) to compete with ATP actively. As shown in Figure 11, the triphosphate of RemTP has other interactions with crucial residues compared to ATP interactions to inhibit NSP12. The adenosine of ATP and the Remdesivir^®^ core also interacted differently from each other in the binding pocket. Additionally, RemTP affected D618, which is an essential residue for the function of SARS-CoV-2 RdRp by forming a hydrogen bond with K798. The binding free energies of ATP and RemTP were represented by FEP calculations, which showed that RemTP binds more strongly than ATP in the pocket of SARS-CoV-2 RdRp with approximately 800 folds in K_d_ value to effectively inhibit SARS-CoV-2 RdRp activity in RNA reproduction [44]. 

In addition, Grein et al. [45] further investigated Remdesivir^®^ clinical application conducted within a cohort of hospitalized patients diagnosed with COVID-19. The data were collected from 53 patients across three continents. The majority of patients (34 out of 53) had severe symptoms and were under invasive oxygen support. The treatment plan was designed for a 10-day course with 200 mg administered intravenously on day 1 and 100 mg daily for the next nine days. Only 40 patients completed ten days of treatments, while others discontinued treatment due to serious adverse effects. After 18 days of day one treatment, 12 patients (100%) previously with no oxygen or low-flow oxygen support were discharged. Five out of seven patients (71%) previously under non-invasive ventilation were also discharged. Nineteen out of thirty-four showed improvement within the oxygen-support group, as they were discharged without oxygen aid and under non-invasive oxygen support. The other patients in this group either showed no improvement (9 out of 34) or died (6 out of 34). About 83% experienced various common to serious adverse effects. Overall, Remdesivir in this study showed improvement in the treatment of COVID-19 based on the oxygen support scale. Among the patients, 68% showed improvement in oxygen supporting status or were fully discharged, with a 13% mortality rate after treatment. 

## 5. Pyrazole

Pyrazole is an aromatic heterocycle composed of a five-membered ring with two nitrogens. Pyrazole core structures have been employed in wide applications from natural components to pharmaceutical activities [46]. With high electron densities of nitrogen at positions 1 and 2, pyrazole is considered as a π-excessive heterocycle. The ring is available for various modifications with multiple reagents in synthetic reactions. Substituted pyrazole can form chelation with metal ions to inhibit enzymes [47].

Currently, HIV-1 is still considered one of the most threatening infectious diseases to people with a high rate of infection. The crucial step of viral infection is the integration of viral RNA into host DNA by reverse transcriptase (RT). RT is composed of a DNA polymerase domain and a ribonuclease H domain (RNase H) [48]. Messore et al. [49] reported the synthesis of specific inhibitory analogs to target RNase H. As shown in Figure 11, the synthesis started with the aldol condensation of substituted benzaldehyde **46** with acetone, then its reaction with TosMIC to furnish pyrrole derivative **47**. Compound **47** coupled with substituted aryl or alkyl chloride for the *N*-substitution of pyrrole ring to form compound **48**. Diethyl oxalate was coupled with intermediate **48** for aldol condensation product **49**. Then, hydrazine reacted with compound **49** for the formation of the pyrazole ring of product **50**. The ester of product **50** was hydrolyzed under a basic condition to yield carboxylic acid for a separate analysis of antiviral activity.

In in vitro screening of RNase H inhibitory activity, compound **50** showed better inhibitory activity of 0.27 ± 0.05 μM compared to its ester version **50** (17.8 ± 1.2). However, compound **50** had a better effective concentration (EC_50_), higher cytotoxicity, and selectivity index; hence, compound **50** was chosen for analysis in the docking study, as shown in Figure 12. The nitrogen of pyrazole ring and ester moiety formed ligands with two magnesium ions in the core of RNase H ligands. The ester moiety also had hydrogen interaction with H539, while the substituted benzyl group was possibly linked with W535 and K540 by π–cation interaction. The phenyl moiety was linked with Y501, S499, Q475, E478, N474, and Q475. The scaffold exhibited high potency with the HIV-1 RT pocket and could be optimized for drug development. 

Annually, the influenza virus causes respiratory infection, known as flu, in millions of people in the United States and worldwide. A vaccine is available to prevent the massive outbreak of the disease, but it does not efficiently protect children and elders from the infection. Hence, inhibitors were continuously developed for a long-term battle against influenza viruses. Among many molecular drug targets of influenza viruses, only neuraminidase (NA) inhibitors were approved for treatment [50,51]. Meng et al. [52] synthesized pyrazole derivatives as neuraminidase inhibitors. As shown in Figure 12, 4-methoxy substituted benzaldehyde **52**, phenylhydrazine **53**, and ethyl acetoacetate **54** were allowed to react in the presence of cerium (IV) ammonium nitrate (CAN) and PEG 400 in a one-pot synthesis. The reaction time and yield were optimized with CAN compared to other catalysts to yield pyrazole derivative **55**.

As shown in Table 7, compound **55** has a high potency to inhibit the replication of viruses with an IC_50_ of 5.4 ± 0.34 μM and a high selectivity index of 102.20 μM in the assay with the influenza A (H1N1) virus. Additionally, the analog also showed great potency (1.32 ± 0.06 μM) against neuraminidase. As shown in Figure 13, the molecular docking of compound **55** in the 3D structure of neuraminidase (NA) was observed. Compound **55** can fit into the binding pocket of neuraminidase and form two hydrogen bonds with residues Arg371 and Arg118 from the methoxy of the compound. The aromatic ring also stabilized with Arg152 through π–cation interaction. With all of the interacted factors in the binding site of NA, compound **55** efficiently exerted inhibitory activity against the targeted protein.

Hepatitis B is another viral infection that is caused by a member of the Hepadnaviridae family. Hepatitis B virus (HBV) has infected millions of people worldwide and the deaths of more than 700,000 people per year. Its replication process is similar to that of RNA retrovirus using the reverse transcription of RNA progenome. Jia et al. [53] developed non-nucleoside HBV inhibitors to combat HBV. As shown in Figure 13, diethyl carbonate **56** coupled with 4-methylpentan-2-one to produce diketone **57**. Compound **57** reacted with 1,1-dimethoxy-*N, N*-dimethylmethanamine for α-substitution to furnish compound **58**. The substituted hydrazine condensed with compound **58** for pyrazole ring formation of compound **59**. Pyrazole derivative **60** was produced via base hydrolysis of compound **59**. The carboxylic acid of **60** continued to react with an aryl-substituted amine under the coupling condition to form amide **61**.

As shown in Table 8, compound **61** demonstrated excellent cytotoxicity in the SAR study. Comparing the inhibitory activity, compound **61** seemed to inhibit viral proteins (HBeAg) of HBV more effectively with a lower IC_50_ (2.22 μM) and corresponding selectivity index (37.69 μM) than the surface antigen (HBsAg). According to the study, compound **61** showed moderate inhibitory activity of approximately 50% against HBV DNA replication. As a result, compound **61** needs further investigation to optimize its inhibitory properties in targeting viral proteins and inhibiting the fusing of viral DNA into host cells. 

## 6. Quinoline

Quinoline is a heterocycle made of benzene and a fused pyridine ring found in shale oil, coal, and petroleum. This heterocycle attracted attention in organic chemistry for its feasible synthesis and availability for substitution modifications. Quinoline derivatives have been employed in various applications, including those in the pharmaceutical, bioorganic, and industrial chemistry fields [54]. Quinoline skeletons exhibited a broad range of antiviral activities and were submitted for potential clinical applications.

With the progress in developing inhibitors at the RNase H of HIV RT, quinoline scaffolds also joined the race to combat HIV-1. While HIV RT synthesizes DNA from viral RNA, RNase H plays a crucial role in degrading RNA to form double-stranded DNA. Overacker et al. [55] synthesized quinoline derivatives to target HIV RNase H. As shown in Figure 14, the methanolysis of carbamate moiety in compound **62** took place to afford quinol derivative **63**, which then underwent Williamson ether synthesis, converting to isopropyl ether to furnish product **64**. 

As shown in Table 9, quinoline derivative **64** required a low concentration for antiviral activity and a high concentration for cytotoxicity in the in vitro infectivity assay with pseudoviruses. The compound also had a high selectivity index to inhibit the growth of HIV. For the mode of action, the compound showed better inhibition against the HIV-1 RNase H enzyme compared to weak inhibition (>100 μM) in HIV-1 integrase and HIV-1 RT. As shown in Figure 14, no changes could be detected from the UV spectra, indicating that compound **64** did not chelate with Mg^2+^ ion in RNase H’s core despite varying concentrations. The structure possibly acted differently in a cell environment or worked as a non-competitive inhibitor of RNase H. This compound was a lead compound for further modification to target RNase H.

Due to the rapid emergence of hepatitis C virus (HCV) drug resistance, Shah et al. [56] reported the synthesis of quinoline derivatives as HCV drug candidates to target the NS3/4a protease, which plays crucial roles in processing viral protein and replicating viral RNA. It also inhibits the production of interferons that can enhance the immune system against viral infection [57]. The new scaffold was developed based on previous clinical candidate **MK-5172** (Figure 15), with a quinoline moiety instead of quinoxaline. As shown in Figure 15, compound **65** was synthesized by linking the amine derivative with quinoline via five steps. The hydroxyl group was protected using *N*-Boc-4-hydroxypiperidine before the methyl ester was hydrolyzed and coupled with an amine-acyl sulfonamide chain. The *N*-Boc-piperidine was then deprotected for the addition of bicyclic piperidine to produce product **66**. 

Compound **66** exhibited improved potency against HCV genotype 3a and A165 mutants compared to reference MK-5172 and had a moderate pharmacokinetic profile in rats in the SAR study. Compared to MK-5172, the replacement of quinoxaline for quinoline in compound **66** enhanced its interaction with the binding pocket HCV NS3 protease. As shown in Figure 15, the substitution at C-4 of quinoline mainly improved the interaction with D79. The replacement of t-butyl with cyclohexane near P3 or modifying cyclopropyl group seemed to not impact the activity of the scaffold in the HCV NS3 protease binding pocket. 

Wang et al. [58] reported the synthesis of indoloquinoline or quindoline derivatives as potential anti-influenza A agents to counter drug resistance of viral strains. As shown in Figure 16, the 11-chloroquindoline **67** reacted with benzene-1,2-diamine to produce compound **68**. In the study, benzene-1,2-diamine could be changed to 1,3 and 1,4-diamine, which then further affected the position of carboxyphenylboronic acid for different analogs of the scaffold. The final product, compound **69,** was produced by the coupling reaction of compound **68** and 2-carboxyphenylboronic acid.

Compound **69** exhibited superior properties compared to other derivatives in this scaffold with low cytotoxicity. As shown in Figure 16, the plaque reduction assay showed that plaque development was effectively inhibited, indicating that the compound could inhibit the replication of influenza H1N1 and H3N1. In the confocal imaging, compound **69** was detected in the cytoplasm to target viral neuraminidase and prevent the import of viruses. Compound **69** could interfere with cellular signaling pathways that are essential for viral replication and improve the survival rate of the mouse model in a histopathology study. The quindoline with boronic acid possessed broad anti-influenza A activity for future references and applications.

In the early study of COVID-19, besides Remdesivir^®^, Chloroquine^®^, an anti-malarial drug (Figure 17), exhibited antiviral activity against 2019-n-CoVs infection in vitro. As shown in Figure 18, Chloroquine^®^ could effectively inhibit viral infection at a low concentration and with good cytotoxicity compared to other FDA-approved antiviral agents. Additionally, Chloroquine^®^ also effectively reduced viral copies with 10 μM after 48 h post-infection in the nucleoprotein through immunofluorescence assay [43]. 

As shown in Figure 18, Chloroquine showed its effectiveness in blocking virus infection at EC_50_ = 1.13 µM concentration with a high selectivity index (SI). The immunofluorescence assay also confirmed the efficiency of Chloroquine in inhibiting viruses, as the viral nucleoproteins were eliminated completely with 10 µM. Together with Chloroquine^®^, Hydroxychloroquine^®^ was also tested for its anti-2019-nCoV activity. Hydroxychloroquine possessed less toxicity in cells compared to Chloroquine^®^ with anti-inflammatory properties [59]. However, both Chloroquine^®^ and Hydroxychloroquine^®^ have side effects, interfering with lysosomal activity and causing cardiac and skeletal muscle problems [60]. 

## 7. Conclusions

Nitrogen-containing heterocycle derivatives have a long history against a broad spectrum of viral agents. Their structures are similar to the biological components that are vital for the infection mechanism and replication of viruses. Incorporating nitrogen-containing heterocycles enhances the binding affinity of the scaffolds by increasing interaction with residues and matching with the pocket shape to deactivate the function of targeted enzymes. This article summarized the history, synthesis, and antiviral applications of various nitrogen-containing heterocycle scaffolds, such as indoles, pyrroles, pyrimidines, pyrazoles, and quinolines. The scaffolds were applied to a variety of viruses, ranging from HIV and HCV/HBV to VZV/HSV, SARS-CoV, and influenza viruses. Some approved scaffolds exhibit multiple functions in targeting different viruses, similar to applying Remdesivir^®^, Chloroquine^®,^ and Hydrochloroquine^®^ for the recent SARS-Co-V 2 viruses. Most scaffolds exhibited potential antiviral activity that could lead to further optimization and in vivo studies for drug development. For example, for HCV treatment, FDA-approved drugs such as Grazoprevir, Voxilaprevir, and Glecaprevir were modified from scaffold **66**. Viruses have high rates of emergence that demand continual drug discovery of new antiviral agents. Research on nitrogen heterocycles will progressively expand for future drug development.

## Data Availability

The data presented in this study are available on request from the corresponding author.

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
