# Peer review of "Synthesis and Applications of Nitrogen-Containing Heterocycles as Antiviral Agents"

_molecules, 2022, doi:10.3390/molecules27092700_

Round 1

Reviewer 1 Report

The manuscript entitled "Synthesis and Applications of Nitrogen-Containing Heterocycles as 2 Antiviral Agents” presents a detail review of the bioactivity and inhibitory properties of the heterocyclic compounds. The study presents comprehensive information in the field of pharmacology. The presentation of the review is good.

I have the following comments to make to enhance the quality of the paper:

  1. Carefully revise the English for typos and errors through the manuscript
  2. Introduction could be improved giving more emphasis to the role of organic compounds as enzyme inhibitory effects and biologically active compounds, at this regard look at the following studies and referred them.

“An in vitro and in silico study on the synthesis and characterization of novel bis (sulfonate) derivatives as tyrosinase and pancreatic lipase inhibitors”

 “2-methylindole analogs as cholinesterases and glutathione S-transferase inhibitors: Synthesis, biological evaluation, molecular docking, and pharmacokinetic studies”

 “Synthesis, Characterization, Enzyme Inhibitory Activity, and Molecular Docking Analysis of a New Series of Thiophene-Based Heterocyclic Compounds”

Overall my recommendation is minor after changes

Author Response

Response to Reviewer 1

The manuscript entitled "Synthesis and Applications of Nitrogen-Containing Heterocycles as Antiviral Agents” presents a detail review of the bioactivity and inhibitory properties of the heterocyclic compounds. The study presents comprehensive information in the field of pharmacology. The presentation of the review is good.

I have the following comments to make to enhance the quality of the paper:

The authors would like to thank Reviewer 1 for the informative comments to help us to improve our review article. The manuscript was updated as shown below.

  1. Carefully revise the English for typos and errors through the manuscript

- The typos and errors have carefully revised to improve the content and accommodate broad readership.

  1. Introduction could be improved giving more emphasis to the role of organic compounds as enzyme inhibitory effects and biologically active compounds, at this regard look at the following studies and referred them.

“An in vitro and in silico study on the synthesis and characterization of novel bis (sulfonate) derivatives as tyrosinase and pancreatic lipase inhibitors”, see reference # [5]

“2-methylindole analogs as cholinesterases and glutathione S-transferase inhibitors: Synthesis, biological evaluation, molecular docking, and pharmacokinetic studies”, see reference # [6].

 “Synthesis, Characterization, Enzyme Inhibitory Activity, and Molecular Docking Analysis of a New Series of Thiophene-Based Heterocyclic Compounds”.

Overall my recommendation is minor after changes

We thank reviewer 1 for the constructive comment. The introduction was improved with emphasizing the advantages of heterocycles with different interaction for ligand binding in the pocket of targeted enzymes. The recommended studies provided helpful information about the roles of heterocycles and were cited in the below sentences in the Introduction of manuscript.

“Those heterocyclic backbones have rigid aromatic structures that can be incorporated into the binding pockets and provide various molecular interactions, such as ionic bonding, hydrogen bonding, hydrophobic interaction, non-covalent bonding, etc., for ligand binding with receptor proteins [5,6]”. (line 57-60)

Reviewer 2 Report

The manuscript “Synthesis and Applications of Nitrogen-Containing Heterocycles as Antiviral Agents” is devoted to a relevant topic and, thus, is interesting for the broad readership of Molecules. In my opinion, the paper can be accepted after improvement of the introduction section.

  1. The introduction should contain information on previous reviews on this topic or close ones.
  2. The introduction should clearly indicate the subject of the review and explain the choice of it. So, the choice of indoles, pyrroles, pyrimidines, pyrazoles and quinolines should be explained in more details.
  3. The introduction should explain the time period of the literature covered in the review.
  4. The phrase “capable of ligand binding with receptors” (line 52) is not clear.

Author Response

Response to Reviewer 2

The manuscript “Synthesis and Applications of Nitrogen-Containing Heterocycles as Antiviral Agents” is devoted to a relevant topic and, thus, is interesting for the broad readership of Molecules. In my opinion, the paper can be accepted after improvement of the introduction section.

The author sincerely appreciated the intricate comments from Reviewer 2 to revise the manuscript. The introduction has been revised with some background on previous reviews on similar topic and detail on heterocycles as addressing below.

  1. The introduction should contain information on previous reviews on this topic or close ones.

Information on previous review were introduced to our introduction of manuscript to differentiate between the purpose of our review and the previous ones.

“Previous reviews focused on one type of nitrogen-containing heterocycles such as indole and/or imidazothiazole derivatives in designing antiviral agents [9,10]. Other reviews covered broader details on various biological applications such as antimicrobial, anti-inflammatory, anti-tubercular, anti-depressant, and anti-cancer activities [11-13].” (line 63-66)

  1. The introduction should clearly indicate the subject of the review and explain the choice of it. So, the choice of indoles, pyrroles, pyrimidines, pyrazoles and quinolines should be explained in more details.

The detailed advantages of heterocycles were expanded as shown below in the Introduction to explain the choice of heterocycles for our review.

“While the indole cores were known for their wide existence in natural products with biological activity [9], the properties of pyrrole can be expanded for chemical design and are suitable for biological systems [14]. Pyrimidine derivatives have widespread therapeutic applications as they are essential building blocks of nucleic acids in DNA and RNA [15]. Pyrazoles can be fused with other heterocycles to extend their biological active potentials [16]. Quinoline derivatives have versatile chemical properties for synthesis and biological activities [17].” (line 71-77)

  1. The introduction should explain the time period of the literature covered in the review.

The below sentence is added to address the time period of the literature covered in the manuscript.

“Our current review focused specifically on the common nitrogen-containing heterocycles including indoles, pyrroles, pyrimidines, pyrazoles, and quinolines that have been applied in drug design for antiviral purposes during the past ten years.” (line 66-69)

  1. The phrase “capable of ligand binding with receptors” (line 52) is not clear.

The sentence was expanded as below to make the phrase more comprehensible with additional information. The additional sentence will support the claim on the advantage of heterocycles in ligand binding.

“Those heterocyclic backbones have rigid aromatic structures that can be incorporated into the binding pockets and provide various molecular interactions, such as ionic bonding, hydrogen bonding, hydrophobic interaction, non-covalent bonding, etc., for ligand binding with receptor proteins [5,6].” (line 57-60)

Reviewer 3 Report

With joy I read the review entitled "Synthesis and Applications of Nitrogen-Containing Heterocycles as Antiviral Agents" by Tran and Henary. It is a meticulous and comprehensive work, facilitating the reader to know the state of the art. I consider the work is publishable in Molecules without any changes in the main text.

However, there are some minor points that the authors should address in any subsequent revision:

  1. Reference 3 has a different format than other references (regarding DOI).
  2. The authors should increase the quality of the figures because some of them look pixelated.

Author Response

Response to Reviewer 3

Comments and Suggestions for Authors

With joy I read the review entitled "Synthesis and Applications of Nitrogen-Containing Heterocycles as Antiviral Agents" by Tran and Henary. It is a meticulous and comprehensive work, facilitating the reader to know the state of the art. I consider the work is publishable in Molecules without any changes in the main text.

The authors would like to thank the supportive comments from Reviewer 3 to help us enhance the manuscript before publication.

However, there are some minor points that the authors should address in any subsequent revision:

  1. Reference 3 has a different format than other references (regarding DOI).

The authors have updated the format of reference 3 to match with other references. The author also checked the rest of references to ensure they have the same format.

  1. The authors should increase the quality of the figures because some of them look pixelated.

The authors have increased the quality of most figures in the manuscripts. Those figures now have overall improved quality compared to before.

We feel that our manuscript would fit for Molecules and it would motivate many chemists and other experts to explore the Nitrogen-Containing Heterocycles as Antiviral Agents. If you have any questions or concerns, please feel free to contact me at [email protected] or call me at (404) 413-5566. We are looking forward to hearing back from you regarding our review.

Round 2

Reviewer 2 Report

The authors have made necessary changes to the manuscript. Now it is ready for publication.